# Diffraction Efficiency of MEMS Phase Light Modulator, TI-PLM, for Quasi-Continuous and Multi-Point Beam Steering

**DOI:** 10.3390/mi13060966

**Published:** 2022-06-18

**Authors:** Xianyue Deng, Chin-I Tang, Chuan Luo, Yuzuru Takashima

**Affiliations:** James C. Wyant College of Optical Science, University of Arizona, Tucson, AZ 85719, USA; chinitang@email.arizona.edu (C.-I.T.); chuanluo@email.arizona.edu (C.L.)

**Keywords:** LiDAR, phase light modulator (PLM), MEMS, computer-generated hologram (CGH), GS algorithm, beam steering

## Abstract

The recent development of the Micro Electromechanical System (MEMS) Phase Light Modulator (PLM) enables fast laser beam steering for lidar applications by displaying a Computer-Generated Hologram (CGH) without employing an iterative CGH calculation algorithm. We discuss the application of MEMS PLM (Texas Instruments PLM) for quasi-continuous laser beam steering by deterministically calculated CGHs. The effect on the diffraction efficiency of PLM non-equally spaced phase levels was quantified. We also address the CGH calculation algorithm and an experimental demonstration that steered and scanned the beam into multiple regions of interest points, enabling beam steering for lidar without sequential raster scanning.

## 1. Introduction

Laser beam steering is essential for applications such as lidar, display, metrology, optical interconnects [1,2], and object tracking [3] as well as scientific research tools, such as optical tweezers [4,5]. Among these, the lidar application requires simultaneously achieving mutually conflicting requirements, such as a high photon throughput, a fast scanning rate, a large angular throw, and a large device area, to detect objects at long distances with a high frame rate while ensuring laser safety for the eye [6]. These requirements are often conflicting. For example, a mechanically rotating scanner can be used for beam steering with a large beam size, but such high-inertia mechanics limit the scan rate [7]. At the other end of the spectrum, non-mechanical laser beam steering approaches, such as Liquid Crystal on Silicon (LCoS) spatial light modulators (SLMs), have been adopted for high-efficiency beam steering [8]. However, the slow response time of liquid crystals on the order of ms limits the speed of scanning to sub-kHz [7]. The linear polarization required for LCoS SLMs reduces the photon throughput by half for a high-power unpolarized laser source. Micro Electromechanical System (MEMS)-based resonant mirrors can achieve an impressive angular throw of over several tens of degrees and a fast-axis resonant frequency with a refresh rate of tens of kHz, but the small mirror aperture size limits the output pulse energy, limiting the maximum range while satisfying the requirement for laser safety [9].

Arrayed MEMS SLMs are uniquely positioned to satisfy such conflicting requirements. Commercially available MEMS SLMs, such as the Digital Micromirror Device (DMD), accommodate an array area of over 140 mm^2^ [10]. Over 48 degrees of angular throw by diffractive beam steering is demonstrated by employing short-pulse illumination in a synchronous manner to the movement of the DMD [11]. The combination of two scanning modalities by employing pulsed illumination increases the number of scanning points without sacrificing the fast refresh rate of the MEMS SLM [12]. As these works show large etendue (the product of area and angular throw) of arrayed MEMS SLMs, laser beam steering is feasible because of the wide field of view, increased range for object detection, and lower laser power density, which satisfies eye safety regulations.

Diffraction efficiency is one of the key metrics of arrayed MEMS SLMs for beam steering. Recently, a MEMS-based phase SLM, Texas Instruments Phase Light Modulator (TI-PLM), became available [13]. The TI-PLM modulates phase in piston mode. The micromirror’s height is discretely controlled by a combination of voltage levels applied to electrodes underneath the mirrors. In this manner, the phase is modulated at 16 levels (=3 bits), including the zero level. Figure 1a,b shows the structure of the PLM micromirror. Since the voltage levels applied for three electrodes are usually the same, the electrostatic force between the mirror and electrode primarily depends on the selection and combination of electrodes to which actuating voltage is applied. Consequently, the phase levels are not equally distributed across the 0 to 2 p phase, but they are non-linearly distributed. Consequently, such phase quantization error diffraction efficiency and the decrease in diffraction efficiency are CGH-specific.

In beam steering applications for lidar, adaptive beam steering is more desirable compared with raster scanning. For example, beam steering in automotive time-of-flight lidar applications requires a high frame rate (several tens of frames) with a wide field of view (over tens of degrees) and high angular resolution (0.5 degrees or less). The frame rate of raster scanning-based detection is limited by the data processing speed and/or the traveling time of light. Typically, the sampling rate/degree/channel has about the same value across different scanning modalities: 50–60 samples/s/degree/channel. Due to the limited sampling rate, the frame rate decreases as the FOV and angular resolution increase. One possible option to work around this is adaptive beam steering. Instead of steering beam in raster scan mode, beams are delivered to multiple N objects using prior knowledge, such as camera and radar images. In this manner, a wide FOV and a high angular resolution are simultaneously achieved without decreasing the frame rate. Programable MEMS SLMs are suitable for this scenario.

To achieve adaptive and flexible beam steering for lidar applications, in this paper, we analyze the diffraction efficiency of CGH for quasi-continuous beam steering under the limited number of available phase levels of TI-PLM. In Section 2, we analyze the effects of the non-linearity of the available phase levels of TI-PLM on diffraction efficiency. We correlate the diffraction efficiency to the RMS phase error of CGH, which provides an effective way to improve diffraction efficiency in quasi-continuous beam steering. In Section 3, we demonstrate the algorithm to steer multiple beams using deterministically calculated CGHs. In Section 4, the difference between the diffraction efficiency of single- and two-point beam steering is addressed.

## 2. Effect of Non-Linearly Distributed Phase Levels for Quasi-Continuous and Single Point Beam Steering

### Diffraction Efficiency of TI-PLM

For beam steering, a staircase-approximated sawtooth-shaped (blazed) phase is commonly used to maximize the diffraction efficiency. The diffraction efficiency for the *n*-th diffraction order $\eta_{n}$ is given by [14],
(1)$$\;\;\;\;\;\;\;\;\eta_{n}={\left(\frac{sinc\left(\frac{n}{m}\right)sinc\left(\frac{\phi_{0}}{2\pi }-n\right)}{sinc\left[\frac{1}{m}\left(\frac{\phi_{0}}{2\pi }-n\right)\right]}\right)}^{2}$$
where m is the number of the discretization levels; $\phi_{0}$ is the maximum phase modulation, for example, 2π and n are the diffraction orders [15]. According to Equation (1), the diffraction efficiency of the first-order diffraction approaches 100% as $\phi_{0}$ and m approach 2π and infinity, respectively. However, for beam steering using pixelated MEMS PLM, this is not the case. First, the number of phase levels is limited, i.e., 16 levels for TI-PLM. In addition, the phase levels are not equally distributed across the range of 0 to 2π but are distributed rather non-uniformly. In Figure 2a, the solid line plots the mechanical displacement of MEMS pixels of sixteen phase levels [16]. The maximum displacement of 100% modulates phase in 2π at a wavelength of 633 nm. Compared with the equally distributed phase levels (the dotted line), phase modulation depths between 20 and 40% are not available. This is due to the limited degree of freedom in controlling electrostatic force between the MEMS mirror and the electrodes. Aside from the non-equally distributed phase levels, the period of the CGH for beam steering is not necessarily an integer multiple of physical pixel period D (i.e., 10.8 um for TI-PLM), but it is rather a non-integer multiple of *D*, *rD*, where r≥2 is any real number to steer the beam in a quasi-continuous manner across the field of view.

Upon the calculation of the CGH, the 16 non-uniformly-spaced phase levels were mapped to staircase-approximated sawtooth profile using the following procedure (Figure 2b). A sawtooth blazed phase profile with a periodicity of *L,* and phase modulation of 100% (=2π) was sampled at an interval of *rD*. The sampled phase levels (depicted in red dotted line) were forced to be fit to the nearest available phase levels of TI-PLM (blue solid line) depicted in Figure 2b. For the case of the non-integer period of *L*, Fourier series expansion was carried out over the period of the integer multiple base periods of $L^{\prime }=LCM\left(L^{\prime },\;L\right)\times L$, where LCM(x, y) is a least common multiplier of values x and y.

The effect of non-linearity in phase levels with non-integer periodicity of CGH on diffraction efficiency was simulated. In Figure 3, the diffraction efficiency (DE) of the CGH is plotted as a function of the periodicity of *r*, where *r* is defined by the CGH period (*L*) divided by the pixel period (*D*), or *L*/*D*. For comparison purposes, three cases were examined, such as (a) the continuous sawtooth phase, (b) phase levels forced to be equally distributed into 16 phase levels, and (c) phase levels forced to be non-equally distributed into 16 phase levels. DE is included in the plot. The diffraction efficiencies for (b) and (c) were numerically calculated by taking a magnitude square of the Fourier coefficients phase profile corresponding to the first diffraction order [17]. 

The DE of the TI-PLM was experimentally evaluated using a collimated 10 mW CW DPSS laser at 532 nm (DJ532-10, Thorlabs, NJ, USA) illuminating a TI-PLM (0.47-inch Texas Instruments PLM, TX, USA) with 960 × 540 pixels with a pixel pitch of D = 10.8 um, which is shown in Figure 4. The beam size of the illumination underfilled the area of the MEMS PLM pixel array region of 10.4 by 5.8 mm. The CGH pattern was calculated using the aforementioned method. The CGH pattern was Fourier transformed by an f = 300 mm lens. A power meter (1918-R, Newport, CA, USA) measured the diffracted power to the first order, and the DE was calculated by normalizing the diffracted power to the input power to the TI-PLM. Note that due to the random tilt and displacement of the micromirror array, the zero-order DE of TI-PLM with a flat phase used for the evaluation was about 65%. Consequently, the measured DE was about 65% of the calculated DE [13].

Figure 3 shows that the simulated and measured diffraction efficiency substantially decreased at specific periodicity *r* = 3, 5.5, 6.5, 7.5, and so on. The drop in the DE was correlated with the deviation of the phase profile, i.e., the phase profile forced to TI-PLM’s available levels deviated from an ideal sawtooth phase profile. The standard deviation of phase $\sigma_{rms}$ is expressed by:(2)$$\sigma_{rms}={\left\{\frac{1}{ND}\overset{N-1}{\underset{p=0}{\sum }}\int_{p\Delta \chi_{p}}^{\left(p+1\right)\Delta \chi_{p}}{\left[\varphi \left(x\right)-\theta \left(p\right)\right]}^{2}dx\right\}}^{1/2}\;\;\;\;\;$$
where φ(x) and θ(p) denote the sawtooth phase profile and the staircase-approximated phase profile for which phase levels were forced to the realizable phase levels depicted in Figure 2; N is the number of pixels within a single complete period of CGH; and D is the physical dimension of the pixels, i.e., 10.8 mm. Figure 5a plots the Strehl intensity ratio ($SIR=1-{\left(\frac{2\pi \sigma_{rms}}{\lambda }\right)}^{2}$) of the non-equal phase levels of CGH (Figure 3, case c) and the calculated ratio of the DE of the PLM-forced phase levels of CGH to the DE of the sawtooth phase of CGH. The proximity of the two data points shows that the DE of CGH is well-correlated with the standard deviation of the phase values via SIR.

Figure 5a shows that in period *r* = 3, DE suffered from a substantial decrease due to the increase in $\sigma_{rms}$. One of the mitigation strategies to recover the DE is applying a small in-plane rotation of CGH during the calculation process while keeping the periodicity *r*. We experimentally evaluated the diffraction efficiency of CGH with *r* = 3 rotated by angles of 0, 0.01, 0.05, 0.1, and 1 degree (Figure 5b). As Figure 5a shows, diffraction efficiency was correlated with the RMS phase error. At the specific period *r* = 3, the diffraction efficiency was recovered from 0.37 to 0.42 by rotating the CGH by a fraction of a degree with respect to the optical axis. The recovery mechanism can be explained as follows: the small rotation of the coordinate basis while generating the CGH enables a better representation of the desired phase profile depicted in Figure 2a. The coordinate rotation employed here and the adjustment of the bias voltage that alters the traveling range of the micromirror potentially improve the diffraction efficiency, particularly for grating period *r*, which suffers from lower diffraction efficiency, provided that a better phase profile (or a profile with a smaller RMS phase error) in the vicinity of the initial phase profile is available.

## 3. Quasi-Continuous and Two-Point Beam Steering

### 3.1. Quasi-Continuous and Two-Point Beam Steering by Using Binary Gratings

Iterative algorithms, such as the Gerchberg–Saxton phase retrieval algorithm, calculate the CGH with an arbitrary number of points with a variable power ratio [18]. Multi-point beam steering benefits the frame rate by avoiding time-consuming raster scanning of objects. However, computational time prohibits such an iterative CGH calculation algorithm from being used for beam steering in real-time applications, such as automotive lidars. Here, we experimentally evaluated the DE of two completely deterministic CGH calculation algorithms to steer the beam into two regions of interest. We first describe the CGH calculation algorithm for two points using binary π-phase grating, followed by the addition of complex fields. Figure 6a,b schematically depicts the first CGH calculation methods using binary π-phase grating. Consider two steering points within the maximum angular range of TI-PLM, **λ**/*D*, where **λ** and *D* are the wavelength and pixel period of MEMS PLM, respectively. To generate two arbitrary points within the angular range, a binary π-phase grating with +1 and −1 orders with an angular separation of *p* = l/(*r**D*) is rotated, and phase tilt is added in the amount of (*q*, *r*). The rotation and phase tilt are given by:(3)$$\theta =\tan^{-1}\frac{\left|\xi_{1}-\eta_{1}\right|}{\left|\xi_{-1}-\eta_{-1}\right|}$$
(4)$$\;\left(q,r\right)=\left(\frac{\left(\xi_{1}+\xi_{-1}\right)}{2},\;\frac{\left(\eta_{1}+\eta_{-1}\right)}{2}\right)$$
where $\left(\xi_{1},\;\eta_{1}\right)$ and $\left(\xi_{-1},\;\eta_{-1}\right)$ are the diffraction angles of the two beams associated with the +1, and −1 diffraction orders, respectively.

The experimental results of two-point beam steering by rotation and the linear shift along the x- and y-directions are depicted in Figure 7a,b, and c respectively.

The combination of separation (*p*), rotation (θ), and the shifting of two points (*q*, *r*) can steer the beam into two ROIs. Figure 8 shows a long-exposure image of beam steering into two ROIs. In each of the frames of the MEMS PLM, the beam is steered into a point of each of the regions. Two ROIs were simultaneously and sequentially scanned updating parameters (*p*, θ, *q*, *r*).

### 3.2. Quasi-Continuous and Two-Point Beam Steering by Complex Field Addition

Alternatively, the CGH is calculated by displaying a phase of complex addition of two fields that generates a two-point steering pattern and is given by,
(5)$$\arg\left(\;e^{j\phi_{1}}+e^{j\phi_{2}}\right)$$
where $e^{j\phi_{k}}$ represents the complex field in the CGH plane. Figure 9 shows the experimental result of two-point beam steering for two ROIs. In a similar way to beam steering using a binary π phase pattern, each ROI was scanned point by point with the CGH that steers the beam into two points. For example, as shown in Figure 9a, the ROIs were scanned from the top-left to the bottom-right. As Figure 9b, c show, arbitrary ROIs were defined and scanned simultaneously.

## 4. Discussion

For single point and quasi-continuous beam steering, the limited number of available phase levels of TI-PLM, along with zero-order reflectivity of the all-flat state of the mirror array, is one of the dominant causes of the decrease in diffraction efficiency compared with theoretical values. Since beam steering is carried out in two dimensions, we evaluated the effect of shift and rotation of binary CGH on DE for single-point quasi-continuous beam steering. Figure 10a,b plot the diffraction efficiency of CGH as a function of (a) the phase tilt and (b) the angle of rotation added to the binary p phase CGH. As shown in Figure 10, adding tilt and rotation to the binary *π* CGH did not cause a significant difference in diffraction efficiency.

As Figure 8 and Figure 9 show, phase-tilted and -rotated binary π phase CGH and a complex field addition yielded the same pattern for two-point beam steering. To further compare these two algorithms, we compared the DE of each point in the left and right ROIs in Figure 8 and Figure 9 by averaging the DE of each ROI. The data are summarized in Table 1. For the asymmetrically placed ROIs shown in Figure 8b and Figure 9b, the diffraction efficiency was not equally divided into two ROIs. The diffraction efficiencies of the two points as a whole were comparable for the two approaches to steering beams into two ROIs. Both methods provided half of the diffraction efficiency of single-point beam steering without losing significant power to other diffraction orders. The asymmetry in the diffraction efficiency in two or more ROIs can be pre-compensated by adding a weight factor in Equation (5) with the look-up table approach. The calibration factor calculates the location and area of the ROIs so that the CGH is calculated by weighting two fields with the values taken from the look-up table.

## 5. Summary

Array mirror-based MEMS SLMs accommodate a large beam size and, with auxiliaries, improve angular throw. Due to the large etendue and fast refresh rate, the MEMS SLM is one of the promising candidates for laser beam steering in a solid-state manner. We have addressed critical aspects to adopting a reflective MEMS SLM, TI-PLM (Texas Instruments Phase Light Modulator), for high-efficiency and quasi-continuous beam steering by using deterministically calculated Computer-Generated Holograms (CGHs). Analysis and experimental evaluation revealed that the overall diffraction efficiency decreased due to the non-linearity of PLM’s available phase levels. The diffraction efficiency decreased in grating periods, i.e., 3- and 6-pixel periods, and was quantitatively connected to a single metric, the Strehl intensity ratio, which is a function of the RMS phase error of the grating phase profile measured from an ideal phase profile. The degradation of diffraction was recovered by a slight modification of the display coordinate basis of CGHs, i.e., the rotation of the CGH pattern by a fraction of a degree with respect to the optical axis of the PLM.

The quasi-continuous beam steering by the non-iterative and deterministic CGH calculation algorithm was applied to two-point beam steering. The demonstrated simultaneous steering beam into two regions of interest enabled adaptive beam steering, especially for high-frame-rate applications such as automotive lidar, by replacing sequential raster scanning over the entire field of view.

In two- or multi-point beam steering, as a natural extension of the current study, the RMS phase error-based assessment of the CGH profile for beam steering provides information to debug CGHs. For example, as a possible means of improvement in the diffraction efficiency, the rotation and/or slight modification of the overall bias voltage to displace phase levels could be identified by RMS phase error-based assessment without resorting to iterative simulations of diffraction efficiency. Moreover, the low diffraction efficiency at specific grating periods needs to be addressed.

## Figures and Tables

**Figure 1 micromachines-13-00966-f001:**
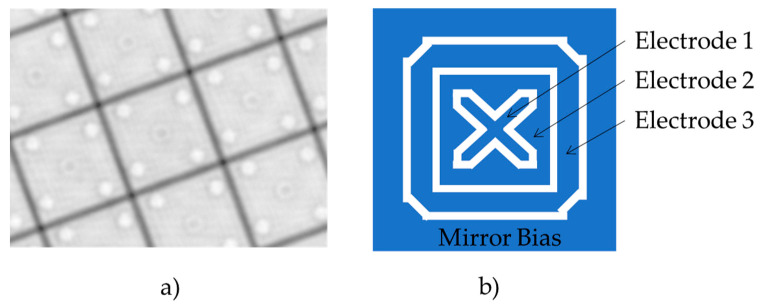
(**a**) White light microscope image of PLM mirror array. (**b**) Schematics of 3-bit PLM micromirror electrostatics electrode arrangement. The three electrodes are placed underneath pixels with 10.8 um pitch.

**Figure 2 micromachines-13-00966-f002:**
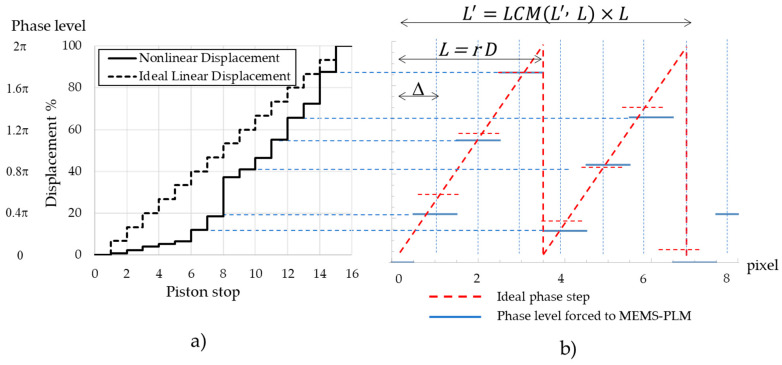
(**a**) Phase profile of 16 equally spaced (dotted line) and realizable phase of TI-PLM (solid line), (**b**) mapping of realizable phase of MEMS PLM to staircase-approximated sawtooth phase profile.

**Figure 3 micromachines-13-00966-f003:**
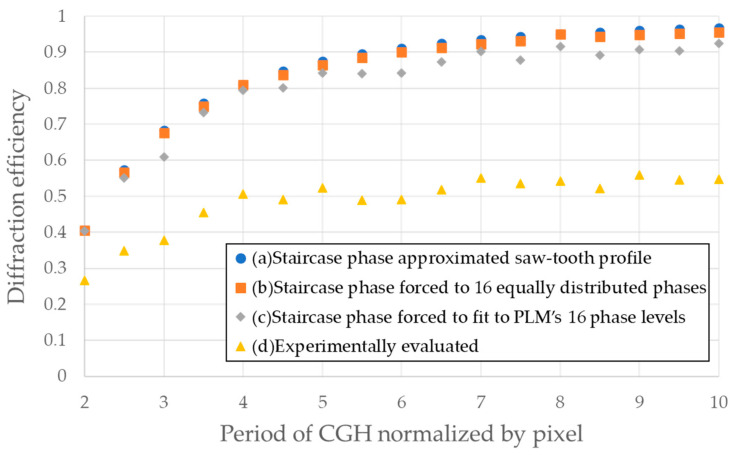
Diffraction efficiency of CGH for (**a**) staircase phase-approximated sawtooth profile, (**b**) staircase phase approximated and forced to fit 16 equally distributed phases across 0 to 2π, (**c**) staircase phase forced to fit TI-PLM’s available 16 phase levels (Figure 2a). and (**d**) experimentally evaluated.

**Figure 4 micromachines-13-00966-f004:**
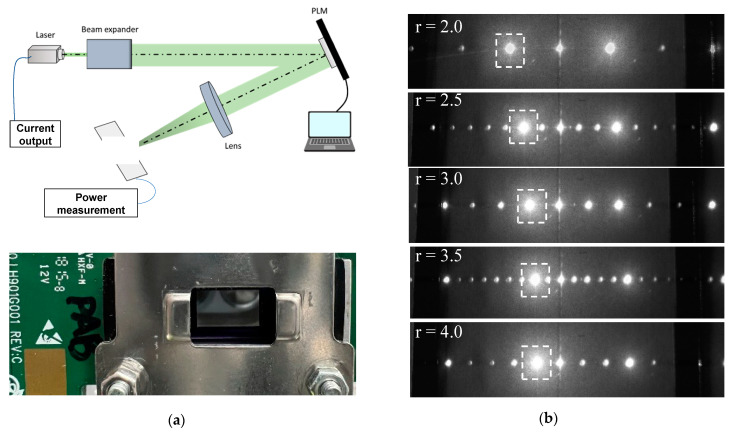
(**a**) Experimental setup. A 532 nm CW laser passing through the beam expander is phase-modulated by TI-PLM, as pictured; (**b**) beam steering pattern for period number *r* = 2.0, 2.5, 3.0, 3.5, and 4.0.

**Figure 5 micromachines-13-00966-f005:**
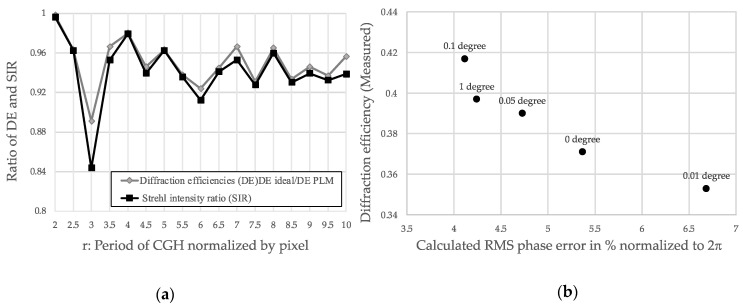
(**a**) Strehl intensity ratio (SIR) and ratio of diffraction efficiency of PLM-forced phase profile to sawtooth phase profile, (**b**) measured DE as a function of $\sigma_{rms}$ for the same period (*r* = 3) of CGH. $\sigma_{rms}$ is controlled by a slight in-plane rotation of CGH.

**Figure 6 micromachines-13-00966-f006:**
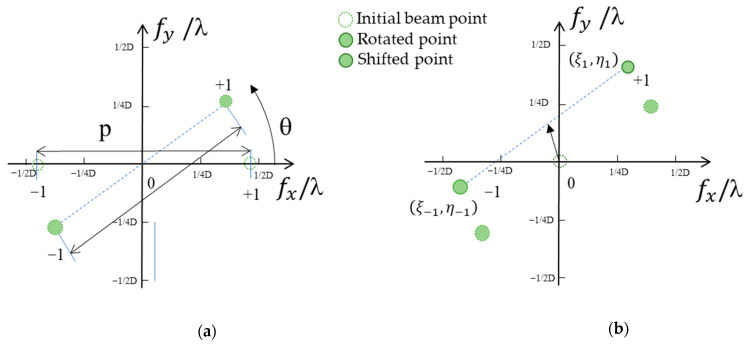
CGH calculation algorithm for beam steering with two points. (**a**) Clockwise rotation. (**b**) Linear shift along the x- and y-directions.

**Figure 7 micromachines-13-00966-f007:**
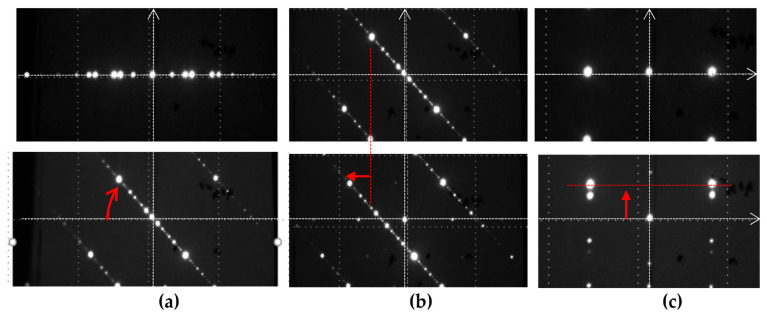
Two-point beam steering: (**a**) rotation, (**b**) linear x-shift, and (**c**) linear y-shift.

**Figure 8 micromachines-13-00966-f008:**
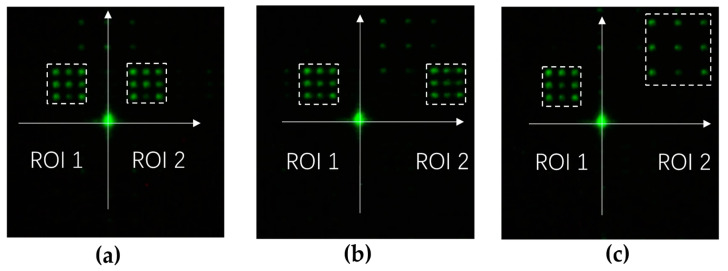
Two-point beam steering using binary gratings with ROIs of (**a**) symmetrical and equal area, (**b**) asymmetrical and equal area, and (**c**) asymmetrical and non-equal area.

**Figure 9 micromachines-13-00966-f009:**
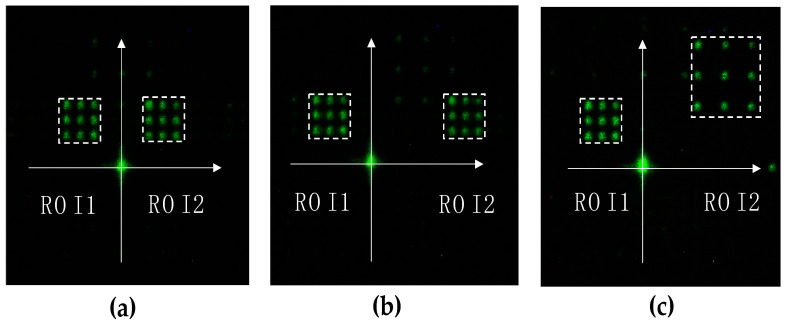
Long exposure of simultaneous scanning of two ROIs. With respect to the *y*-axis, ROIs are defined as (**a**) symmetrical with equal area, (**b**) asymmetrical with equal area, and (**c**) asymmetrical with non-equal area.

**Figure 10 micromachines-13-00966-f010:**
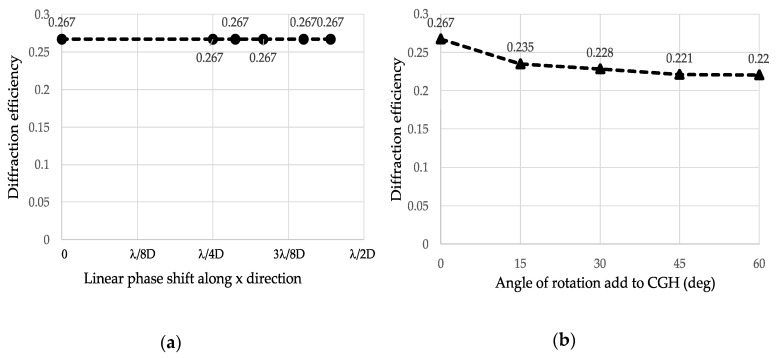
(**a**) Diffraction efficiency of a CGH for single-point beam steering of period number *r* = 2 with horizontal shift, and (**b**) DE for single-point beam steering of period number 2 with rotation.

**Table 1 micromachines-13-00966-t001:** Diffraction efficiency for different algorithms of two-point beam steering.

Pattern	Binary π-Phase Grating			Complex Field Addition		
	ROI 1	ROI 2	Total	ROI 1	ROI 2	Total
(a)	0.1641	0.1629	0.3270	0.1638	0.1600	0.3238
(b)	0.1710	0.1285	0.2995	0.1748	0.1275	0.3023
(c)	0.1641	0.0992	0.2633	0.1609	0.1102	0.2712

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
