# Peer review of "Diffraction Efficiency of MEMS Phase Light Modulator, TI-PLM, for Quasi-Continuous and Multi-Point Beam Steering"

_micromachines, 2022, doi:10.3390/mi13060966_

Round 1

Reviewer 2 Report

The paper "Diffraction efficiency of MEMS phase light modulator, TI-PLM for quasi continuous and multi-point beam steering" present 3 aspects for diffraction efficiency for available phases from a commercially available PLM. 

The three sections are relevant and bring new knowledge to the field, by doing a systematic analysis on nonlinearities, proposing an algorithm to overcome the nonlinearities identified on the first aspect and by analyzing the effect of the algorithm. 

There are some typos, as line 52, and redundancies as in line 24, but otherwise a good written and well-structured article. 

More references to support the motivation would help to strengthen the impact. 

A conclusion section with the interconnection between the three investigated aspects instead of the summary section will also be beneficial. 

Round 2

Reviewer 1 Report

No further comments